# SEEING IT OR NOT? INTERPRETABLE VISION-AWARE LATENT STEERING TO MITIGATE OBJECT HALLUCINATIONS

## ABSTRACT

Large Vision-Language Models (LVLMs) have achieved remarkable success but continue to struggle with object hallucination (OH), generating outputs inconsistent with visual inputs. While previous work has proposed methods to reduce OH, the visual decision-making mechanisms that lead to hallucinations remain poorly understood. In this paper, we propose VaLSe, a Vision-aware Latent Steering framework that adopts an interpretation-then-mitigation strategy to address OH in LVLMs. By tackling dual challenges of modeling complex vision-language interactions and eliminating spurious activation artifacts, VaLSe can generate visual contribution maps that trace how specific visual inputs influence individual output tokens. These maps reveal the model's vision-aware focus regions, which are then used to perform latent space steering, realigning internal representations toward semantically relevant content and reducing hallucinated outputs. Extensive experiments demonstrate that VaLSe is a powerful interpretability tool and an effective method for enhancing model robustness against OH across multiple benchmarks. Furthermore, our analysis uncovers limitations in existing OH evaluation metrics, underscoring the need for more nuanced, interpretable, and visually grounded OH benchmarks in future work.

## 1 INTRODUCTION

Recent advances in large language models (LLMs) (Bai et al., 2023b; Touvron et al., 2023a;b) have accelerated the development of Large Vision-Language Models (LVLMs), such as LLaVA (Liu et al., 2024a; 2023b), InstructBLIP (Dai et al., 2023), MiniGPT-4 (Zhu et al., 2023), and Qwen2-VL (Bai et al., 2023a; Wang et al., 2024). However, LVLMs are prone to object hallucination (Bai et al., 2024; Yang et al., 2025; Duan et al., 2025; Zhou et al., 2024), often generating outputs that are inconsistent with visual inputs, which raises serious concerns about the reliability and safety of LVLMs. Recent efforts to mitigate hallucinations have explored a range of strategies, including end-to-end fine-tuning (Liu et al., 2023a; Jiang et al., 2024; Kim et al., 2023), post-processing of model outputs (Leng et al., 2024; Zhang et al., 2024b; Zhou et al., 2024; Chen et al., 2024c), and latent feature steering (Yang et al., 2025; Chen et al., 2024a; Liu et al., 2025), all of which have shown promising results on open-source LVLMs. Nevertheless, a critical limitation remains (Bai et al., 2024): there still lacks an effective method to trace how visual inputs influence the decision-making processes of LVLMs. As a result, the underlying mechanisms of hallucination and the factors triggering it remain poorly understood.

Interpreting open-ended responses from LVLMs introduces several key challenges. (1) Complex vision-language interaction: The intricate alignment between vision encoders and LLMs creates difficulty in disentangling modality contributions and leads to poor interpretable results (Xing et al., 2025; Stan et al., 2024b). (2) Activation artifacts: Recent studies (Kang et al., 2025; Darcet et al., 2023; Sun et al., 2024a) reveal that some neurons produce disproportionately high activations regardless of the input, which distort visualization results (see Figure 1(a)). These challenges hinder the development of reliable interpretation methods for LVLMs, making it difficult to explore why a hallucinated word is generated and to determine whether a correct prediction is a correct answer or a "guessing one".

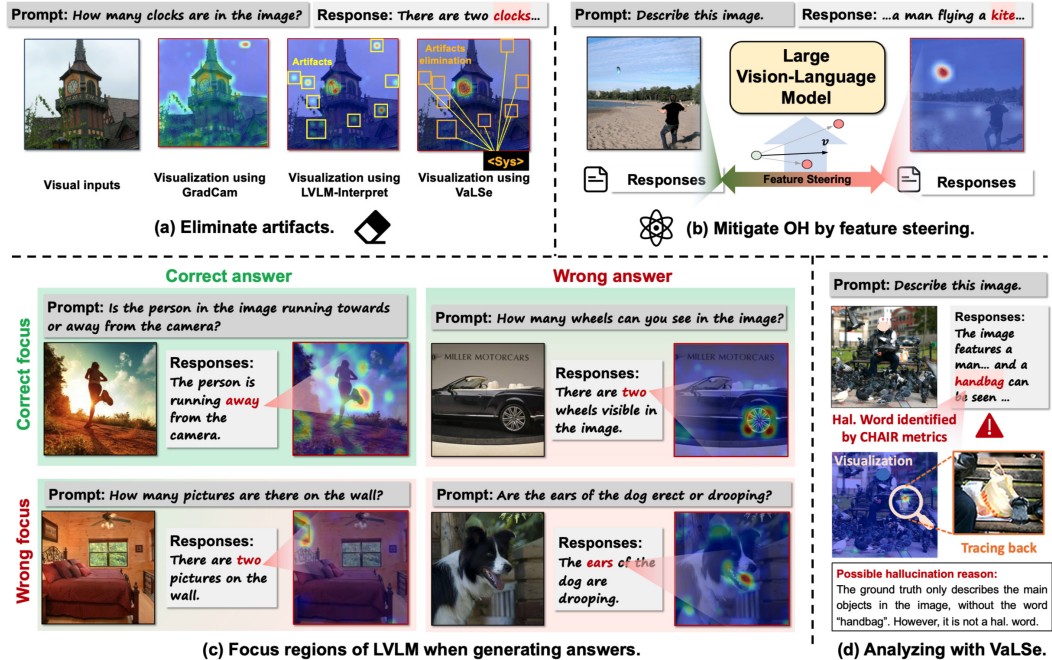

Figure 1: The proposed VaLSe can effectively (a) eliminate artifacts and provide high quality visualization results, and then (b) mitigate OH by vision-aware latent steering. With the ability of mitigating OH, VaLSe can further provide in-depth analysis of (c) how a word token is generated based on visual information and (d) inferring why a hallucinated word is generated.

To address these issues, we propose VaLSe, a novel **V**ision-**a**ware **L**atent **St**eering framework for LVLMs. Figure 1 provides an overview of VaLSe. To trace the influence of visual inputs on output tokens, VaLSe models complex vision-language interactions via visual contribution maps and eliminates artifact activations by contrasting targeted tokens with non-semantic special tokens, resulting in a higher-quality, interpretable visualization of LVLM's output. Moreover, with the interpretable results, VaLSe can reinforce the LVLM's attention to semantically relevant regions in the image by latent feature steering using the generated visual contribution maps, enhancing its visual grounding and effectively mitigating OH (shown in Figure 1 (b)).

Through comprehensive experiments, we demonstrate that VaLSe effectively mitigates OH without compromising general ability. More importantly, VaLSe offers a new perspective for studying OH by providing fine-grained interpretability into the model's decision-making process. As illustrated in the visualization results in Figure 1, which highlight the focus regions of LLaVA-1.5 during response generation, benchmark ground-truth answers alone are insufficient for determining whether hallucination has occurred. On one hand, a model may produce a correct response while attending to irrelevant image regions, indicating it relied on language priors rather than visual cues. For example, in the bottom-left panel of Figure 1 (c), the model correctly predicts the word "two" without attending to the correct relevant visual evidence. On the other hand, as revealed through visualization (Figure 1 (d)), a word flagged as hallucinated by metrics (e.g., the CHAIR metric (Rohrbach et al., 2018)) may actually be a visually grounded and accurate description. These findings highlight not only the importance of understanding the internal mechanisms behind hallucinated outputs but also the need for more sophisticated and comprehensive benchmarks to evaluate OH in LVLMs.

The main contributions are summarized as follows:

- We propose a novel vision-aware latent steering method that follows an interpretation-then-mitigation strategy, enabling internal analysis of the generation process behind hallucinated words and effectively reducing OH in LVLMs.

- VaLSe generates high-quality visual contribution maps across different LVLMs, enabling deeper analysis of their decision-making processes. Our analysis reveals limitations in existing OH evaluation metrics, highlighting the need for more nuanced visually grounded assessment methods.

- Experiments demonstrate the effectiveness of VaLSe in OH mitigation. Moreover, both qualitative and quantitative evaluations demonstrate the superiority of our method in visualization for LVLMs.

## 2 RELATED WORK

**Large Visual-Language Models (LVLMs)** Based on the successes of LLMs, large vision-language models (LVLMs) have made significant progress in recent years. These models typically integrate a vision encoder with an LLM via fusion modules, such as a linear projection layer (Liu et al., 2024a) or a Q-former (Zhu et al., 2023). Recent LVLMs, such as LLaVA (Liu et al., 2024a; 2023b), MiniGPT-4 (Zhu et al., 2023), mPLUG-Owl (Ye et al., 2024b;a), Qwen-VL (Bai et al., 2023b;a), LLaVA-Phi (Zhu et al., 2024b) and DeepSeek-VL (Lu et al., 2024) have been shown to be capable of complex image understanding and reasoning. Despite these advancements, modern LVLMs continue to face significant security and robustness challenges, notably object hallucination (Bai et al., 2024).

**Mitigation of Object Hallucination** Various approaches have been proposed to address this issue. Given that hallucinations may stem from data biases and the knowledge gap between visual and linguistic information, recent studies have explored fine-tuning LVLMs for robustness (Liu et al., 2023a; Gunjal et al., 2024), cross-modality matching (Jiang et al., 2024; Kim et al., 2023), and preference alignment (Sun et al., 2023; Chen et al., 2024b).

To avoid the high cost of fine-tuning, post-processing strategies have been developed to revise model outputs using external tools, such as LURE (Zhou et al., 2024) and visual-guided refiners (Yin et al., 2023; Zhao et al., 2024; Chen et al., 2024c). Other approaches aim to debias strong language priors during decoding (Leng et al., 2024; Liu et al., 2024b; Zhang et al., 2024b; Zhu et al., 2024a; Huang et al., 2024; Favero et al., 2024), while feature-steering methods (Yang et al., 2025; Liu et al., 2025; Fang et al., 2024) learn latent shift directions to adjust internal features for OH mitigation. In contrast, VaLSe not only mitigates OH but also interprets the LVLM's internal generation process, providing insight into the root causes of hallucination. Although ALGA (An et al., 2024) also leverages Grad-CAM to generate saliency-based prompts, it relies on an external multimodal model, making it incapable of explaining the LVLM's own decision-making. VaLSe, by contrast, operates entirely within the LVLM and utilizes its interpretability to directly and effectively reduce OH.

**Interpretation of LVLM.** Interpreting computer vision algorithms often involves generating heatmaps that highlight the relevance of different image regions to the model's decisions. Classical approaches such as Grad-CAM (Selvaraju et al., 2017) and Grad-CAM++(Chattopadhay et al., 2018) achieve this by combining input feature maps with class-specific gradients from the upper layers of convolutional networks. More recently, transformer interpretability has gained growing attention (Chefer et al., 2021a;b; Aflalo et al., 2022), motivating deeper insights into model behavior for interpreting modern LVLMs (Stan et al., 2024b; Xing et al., 2025; Stan et al., 2024a; Giulivi & Boracchi, 2024; Zhang et al., 2024a; Pan et al., 2023). In contrast to these interpretability techniques Stan et al. (2024b); Xing et al. (2025), our method not only provides clearer visual explanations but also leverages them in a feature steering framework, leading to more accurate and reliable outputs by mitigating object hallucinations.

## 3 METHOD

We first present the preliminaries for our method, then introduce the main components of the proposed VaLSe and, finally, provide a brief discussion of VaLSe.

### 3.1 PRELIMINARIES AND NOTATIONS

Suppose we have an LVLM consisting of an image encoder, an alignment module and an LLM with $L$ layers. In the LLM, the hidden states $\mathbf{h}_l$ at layer $l$ can then be calculated as

$$\boldsymbol{h}_l = \boldsymbol{x}_l + \boldsymbol{a}_l, \text{ where } \boldsymbol{x}_l = \mathbf{W}_l^{\text{out}} \, \sigma \left( \mathbf{W}_l^{\text{in}} \left( \boldsymbol{a}_l + \boldsymbol{h}_{l-1} \right) \right) , \; \boldsymbol{a}_l = \sum_{h=1}^{H} Q_l^h (\mathbf{A}_l^h \mathbf{V}_l^h). \quad (1)$$

Here, $\boldsymbol{a}_l$ and $\boldsymbol{x}_l$ represent the outputs of the multi-head attention (MHA) and the multi-layer perceptron (MLP), respectively. The MLP consists of two linear layers with weights $\mathbf{W}_l^{\text{in}}$ and $\mathbf{W}_l^{\text{out}}$, and an activation function $\sigma$. The attention output $\boldsymbol{a}_l$ is computed by aggregating $H$ attention heads. Each head applies an attention map $\mathbf{A}_l^h$ to its corresponding value matrix $\mathbf{V}_l^h$, followed by a projection using $Q_l^h$. For simplicity, layer normalization is omitted from Eq. 1.

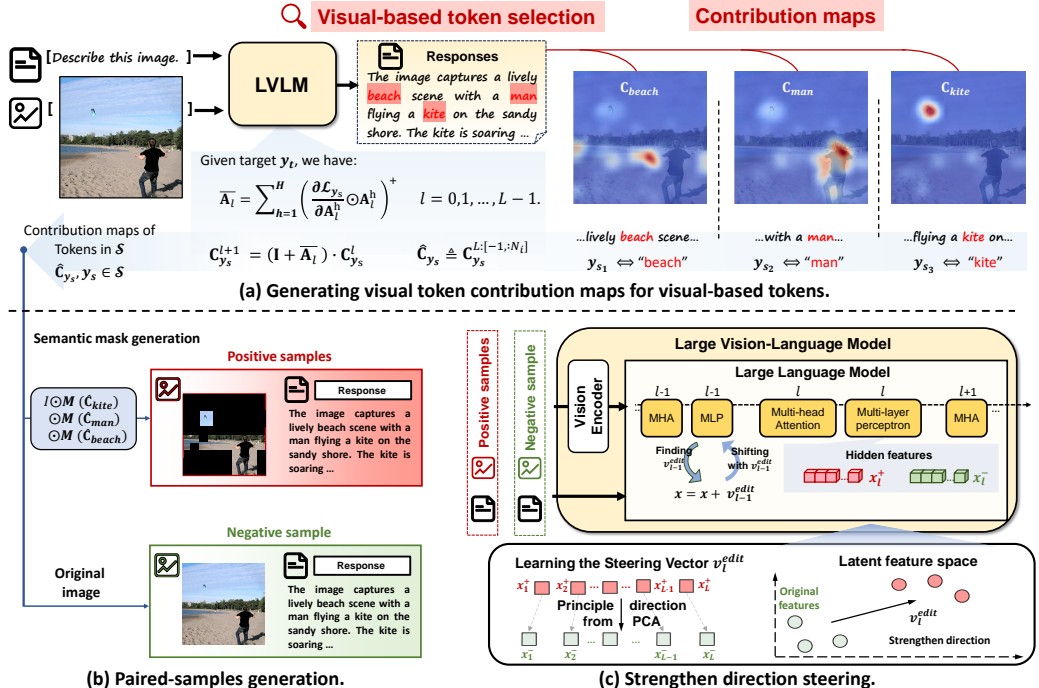

Figure 2: VaLSe mainly contains three modules: (a) A visualization module that generates visual token contribution maps for each selected token; (b) A vision-aware masking module creating masked images while preserving the main semantic contents; (c) A latent steering mechanism.

During autoregressive text generation, words are tokenized and sequentially predicted conditioned on previous tokens. Suppose the answer $y$ consists of $N_r$ tokens, represented as a sequence $y = [y_1, y_2 \cdots y_{N_r}]$. At each step $t$, the model samples the next token $y_t$ according to:

$$y_t \sim P(y_t | y_1, y_2 \cdots y_{t-1}; I, T), \tag{2}$$

where $I$ and $T$ are the input image and text, respectively.

## 3.2 VaLSe

**Overview.** Figure 2 illustrates the main components of VaLSe: (a) Visual-based token selection and contribution map generation, (b) Steering sample construction, and (c) Vision-aware latent steering. The overall procedure is as follows: Given an input image $I$ and a text prompt $T$, the LVLM first generates a response $y$. VaLSe then selects visual-based tokens whose predictions are strongly influenced by visual inputs. For each selected token, VaLSe computes a visual token contribution map, highlighting the image regions the model attends to during token prediction. These maps are then used to construct positive and negative samples for latent steering. The original image and response serve as the negative sample, while positive ones are created by masking obscure, visually irrelevant regions while preserving core vision-aware objects. Finally, VaLSe performs latent steering by computing the directional difference between positive and negative features, adjusting internal representations to reinforce focus on semantically relevant objects and reduce OH.

**Visual-based Token Selection.** A visual-based token is defined as one whose prediction is highly sensitive to the presence of visual information. Following (Xing et al., 2025; Favero et al., 2024), we use the log-likelihood ratio (LLR) between the token's prediction with and without visual context.

Given $I$, $T$, and the generated responses $y_{<t}$, the probability of token $y_t$ is $P(y_t | y_{<t}, I, T)$. To isolate the influence of the image, we can replace $I$ with a noise image $\tilde{I}$ that lacks useful visual information, and compute the probability $P(y_t | y_{<t}, \tilde{I}, T)$. This can be obtained through a single forward pass by concatenating $T$ and $y_{<t}$ as the textual input. The LLR for token $y_t$ is then defined as:

$$\text{LLR}(y_t) = \log P(y_t | y_{<t}, I, T) - \log P(y_t | y_{<t}, \tilde{I}, T). \tag{3}$$

A higher value of $\text{LLR}(y_t)$ represents that the token $y_t$ is generated more highly based on visual inputs. We select tokens with high LLR values, those most influenced by the image. Specifically, we define the set of $S$ visual-sensitive tokens as:

$$\mathcal{S} = \{y_s \mid \text{LLR}(y_s) > \alpha, \ s \neq 1\}, \tag{4}$$

where $\alpha$ is a predefined threshold and $|\mathcal{S}| = S$. The resulting token set $\mathcal{S}$ represents the word tokens in the generated response that are strongly grounded in visual content, which is suitable for visualization[1]. For each of the selected visual-based tokens, we compute the corresponding visual token contribution maps to analyze how the image influences the model's predictions.

**Visual Token Contribution Maps.** Following Chefer et al. (2021a), we compute contribution maps that estimate the relevance of each image token to a specific text token, using the attention mechanisms within the LLM. Let $N_t$[2] and $N_i$ denote the number of text and image tokens, respectively. The attention map at layer $l$ is represented as $\mathbf{A}_l \in \mathbb{R}^{(N_i+N_t) \times (N_i+N_t)}$.

We then generate the visual contribution map $\mathbf{C}_{y_s}$ for $y_s$, which is initialized as an identity matrix and propagated layer-by-layer using $\mathbf{A}_l$. Since each attention layer has $H$ heads, we follow (Chefer et al., 2021b) and compute a weighted average of the heads using their gradients with respect to $y_s$. The aggregated attention map $\bar{\mathbf{A}}_l$ at layer $l$ and propagation of $\mathbf{C}_{y_s}$ can be represented as:

$$\bar{\mathbf{A}}_l = \sum_{h=1}^{H} \left( \frac{\partial \mathcal{L}_{y_s}}{\partial \mathbf{A}_l^h} \odot \mathbf{A}_l^h \right)^+, \quad \mathbf{C}^{l+1} = \mathbf{C}^l + \bar{\mathbf{A}}_l \cdot \mathbf{C}^l, \quad l = 0, 1, ... L-1, \tag{5}$$

where $\odot$ denotes the element-wise product and $(\cdot)^+$ indicates removing negative contributions.

This iterative update propagates relevance scores from the 0-th layer to the $L$-th layer. Since the model typically predicts words based on the last token's hidden state, we take the last row of $\mathbf{C}^L$ and retain the first $N_i$ values, corresponding to the image tokens, $\hat{\mathbf{C}}_{y_s} \triangleq \mathbf{C}_{y_s}^{L[-1,:N_i]}$. Reshaping $\hat{\mathbf{C}}_{y_s}$ yields the visual contribution map for token $y_s$.

**Artifacts Elimination.** Generally, $\hat{\mathbf{C}}_{y_s}$ can be significantly affected by artifact activations, which are neurons that consistently exhibit abnormally high values regardless of the input. These artifacts distort the accurate contribution distribution and compromise interpretability.

Following the observation in (Sun et al., 2024a) that such activations typically occur at fixed spatial positions, we address this issue by contrasting contribution maps between target visual-based tokens and a non-semantic system token $y_{sys}$. Specifically, for $y_{sys}$, we compute its contribution map $\hat{\mathbf{C}}_{sys}$ and identify positions $\mathcal{P}$ exhibiting artifacts. By suppressing these regions in $\hat{\mathbf{C}}_{y_s}$, we obtain cleaner and more accurate visualizations, better reflecting the model's true attention to image content.

**Paired-sample Generation.** For all $N$ samples, we first select $N_s$ vision-aware ones, whose $\mathcal{S}$ is not empty, and mask while preserving key visual information indicated by the selected visual-based tokens for each sample. Specifically, we will generate $S^n$ masks for the $n$-th sample, defined as $\mathcal{M}_n = \left\{ \mathbf{M}(\hat{\mathbf{C}}_{y_s}, \tilde{C}_{y_s}) \mid y_s \in \mathcal{S}^n \right\}$, where $\tilde{C}_{y_s}$ is the mean value of $\hat{\mathbf{C}}_{y_s}$, and $\mathbf{M}(\hat{\mathbf{C}}, \tilde{C}_{y_s})$ denotes the mask obtained by the mean value, which is calculated using different contribution maps. Applying $\mathcal{M}_n$ yields the masked images $\tilde{I}_n = I_n \odot \mathbf{M}(\hat{\mathbf{C}}_{y_1}) \odot \cdots \odot \mathbf{M}(\hat{\mathbf{C}}_{y_{S^n}})$.

The original image $I_n$ and $y$ can constitute the negative sample. Finally, we have $N_s$ negative and positive samples, all of which will be used to perform vision-aware latent steering.

**Vision-aware Latent Steering.** Following Liu et al. (2025), we apply a steering process to the LLM within the LVLM. We first extract the latent states from the MLP layers for both positive and negative samples through forward passes. For the $n$-th sample, let $\boldsymbol{x}_{n,l}^+$ denote the features for the positive samples, and $\boldsymbol{x}_{n,l}^-$ denote the features for the negative samples; these features represent the latent states of the last token in layer $l$ when generating outputs. We compute the direction for each

---

[1]Note that the proposed VaLSe can be used to visualize any token in the response.

[2]$N_t$ includes both the original text prompt tokens and the generated responses.

of the samples as $\Delta_l^n = \boldsymbol{x}_{n,l}^+ - \boldsymbol{x}_{n,l}^-$, then perform PCA on the concatenated directions to extract the overall direction vision-aware directions, $\boldsymbol{v}_l^{\text{edit}}$, consistent with prior studies.

During inference, we apply the learned steering vectors to shift the latent state $\boldsymbol{x}_l$ of LLM by $\tilde{\boldsymbol{x}}_l \leftarrow \boldsymbol{x}_l + \lambda \boldsymbol{v}_l^{\text{edit}}$. Then we normalize the resultant states to the $\ell_2$ norm of the original ones, ensuring that their magnitudes remain consistent with those typically processed by subsequent modules.

$$\tilde{\boldsymbol{x}}_l = \tilde{\boldsymbol{x}}_l \cdot \frac{\|\boldsymbol{x}_l\|_2}{\|\tilde{\boldsymbol{x}}_l\|_2}. \tag{6}$$

### 3.3 WHY VALSE WORKS?

We provide an analysis to understand what the model learns through the latent steering procedure. This analysis can be conducted for each transformer layer $l$; for simplicity, we drop the subscript $l$ and analyze layers independently. Let $f(\boldsymbol{x})$ denote the output of the LVLM given input features $\boldsymbol{x}$, and let $\mathbf{A}$ represent the attention matrix influenced by $\boldsymbol{x}$, $\mathbf{A}(\boldsymbol{x})$. For simplicity, we assume a single attention head. To approximate the model's behavior under perturbed inputs, we apply a first-order Taylor expansion to estimate the output for a noise input $\tilde{\boldsymbol{x}}$, which is expressed as:

$$f(\tilde{\mathbf{A}}) = f(\mathbf{A}) + (\frac{\partial f}{\partial \mathbf{A}})^\top (\tilde{\mathbf{A}} - \mathbf{A}) + \mathcal{R} \Leftrightarrow (\frac{\partial f}{\partial \mathbf{A}})^\top \mathbf{A} = \mathbf{1}^\top (\frac{\partial f}{\partial \mathbf{A}} \odot \mathbf{A}) = \textcolor{blue}{f(\mathbf{A})} - \textcolor{red}{f(\tilde{\mathbf{A}})}, \tag{7}$$

where we suppose all matrices are vectorized and use $\mathbf{A}$ to denote $\mathbf{A}(\boldsymbol{x})$, and $\mathbf{1}$ and $\mathcal{R}$ are the all-one vector and higher-order infinitesimal term, respectively. Since $\tilde{\boldsymbol{x}}$ is assumed to be ideal noise, where tokens are independent of each other, the resulting attention matrix satisfies $\mathbf{A}(\tilde{\boldsymbol{x}}) = \mathbf{0}$.

The blue components in Eq. 7 share the same formulation as the visual contribution maps computed by VaLSe in Eq. 5. Additionally, we observe that the red term closely resembles recent decoding strategies for OH mitigation, such as VCD (Leng et al., 2024) $((1 + \alpha)f(\boldsymbol{x}) - \alpha f(\tilde{\boldsymbol{x}}))$ debiasing the model's prior-driven predictions. Based on this connection, we infer that applying the vision-aware masking via the visual contribution maps enables the resulting latent steering to eliminate model bias at the feature level, similar to the decoding-level as in VCD, and potentially mitigate OH.

## 4 EXPERIMENTS

This section first evaluates the proposed VaLSe in OH mitigation tasks and then conducts a series of visualization experiments to reveal several limitations of existing OH benchmarks. Finally, we further conduct an ablation and analysis experiment.

**Datasets.** We evaluate VaLSe on different popular datasets for hallucination mitigation and general ability evaluation. For OH benchmark, we use CHAIR (Rohrbach et al., 2018), AMBER (Wang et al., 2023), POPE (Li et al., 2023), MMHal (Sun et al., 2024b) and MMVP (Tong et al., 2022) to test the performance of VaLSe in OH mitigation. Moreover, we implement Multi-modal Large Language Model Evaluation benchmark (MME) (Fu et al., 2023), Visual Reasoning and Compositional Question Answering (GQA) (Hudson & Manning, 2019) and LLaVA-Bench (Liu et al., 2023b) to test the general ability of the LVLMs.

**Implementation Details.** To evaluate the effectiveness of VaLSe, we implement VaLSe on three mainstream large vision-language models, including LLaVA-1.5 (Liu et al., 2024a), MiniGPT-4 (Zhu et al., 2023) and Qwen2-VL (Wang et al., 2024). More details are provided in the supplementary materials.

### 4.1 OH MITIGATION RESULTS

**Compared to Existing Methods.** Table 1 summarizes the performance of VaLSe when incorporated into LLaVA-1.5 and MiniGPT-4, in comparison with existing OH mitigation approaches. LLaVA enhanced with VaLSe outperforms all compared methods, while MiniGPT-4 combined with VaLSe achieves performance comparable to most decoding-based baselines. Among the metrics, $C_S$ is particularly critical, as a caption containing multiple correct objects but a single hallucinated one is still considered erroneous. A substantial improvement in $C_S$ indicates that VaLSe effectively eliminates the remaining hallucinated objects. We also report BLEU, F1, and Length (Len) metrics to ensure that VaLSe does not compromise response quality or object coverage. We provide more comparisons in the supplementary materials.

Table 1: CHAIR evaluation results. We use 64 as the max token number in this experiment.

| Method | LLaVA-1.5 | | | | | MiniGPT-4 | | | | |
|---|---|---|---|---|---|---|---|---|---|---|
| | $C_S \downarrow$ | $C_I \downarrow$ | BLEU$\uparrow$ | F1 | Len | $C_S \downarrow$ | $C_I \downarrow$ | BLEU$\uparrow$ | F1 | Len |
| Greedy | $20.4_{\pm 2.8}$ | $7.1_{\pm 0.3}$ | $15.7_{\pm 0.1}$ | 73.2 | 54.7 | $32.4_{\pm 2.2}$ | $12.2_{\pm 0.4}$ | $14.6_{\pm 0.1}$ | 67.9 | 55.4 |
| Beam Search | $19.5_{\pm 2.3}$ | $6.8_{\pm 0.8}$ | $16.0_{\pm 0.1}$ | 71.7 | 50.0 | $30.1_{\pm 0.3}$ | $11.9_{\pm 0.4}$ | $15.4_{\pm 0.2}$ | 67.4 | 54.3 |
| DoLa (Chuang et al., 2023) | $20.2_{\pm 2.8}$ | $6.8_{\pm 0.5}$ | $15.7_{\pm 0.1}$ | 72.5 | 52.1 | $31.9_{\pm 3.3}$ | $12.2_{\pm 0.9}$ | $14.5_{\pm 0.1}$ | 68.1 | 55.8 |
| OPERA (Huang et al., 2024) | $17.5_{\pm 0.5}$ | $6.1_{\pm 0.3}$ | $16.0_{\pm 0.1}$ | 72.6 | 53.1 | $29.7_{\pm 0.3}$ | $12.0_{\pm 0.3}$ | $14.8_{\pm 0.1}$ | 67.1 | 54.6 |
| VCD (Leng et al., 2024) | $20.3_{\pm 1.1}$ | $7.3_{\pm 0.1}$ | $14.5_{\pm 0.0}$ | 71.0 | 51.6 | $29.0_{\pm 2.8}$ | $12.6_{\pm 1.2}$ | $14.4_{\pm 0.0}$ | 66.2 | 53.1 |
| HALC (Chen et al., 2024c) | $16.9_{\pm 2.1}$ | $5.7_{\pm 0.6}$ | $16.0_{\pm 0.1}$ | 71.2 | 51.0 | $\mathbf{25.2}_{\pm 2.0}$ | $\mathbf{9.4}_{\pm 0.4}$ | $14.9_{\pm 0.1}$ | 67.4 | 53.8 |
| VTI-v (Liu et al., 2025) | $17.4_{\pm 2.0}$ | $6.0_{\pm 0.6}$ | $15.5_{\pm 0.1}$ | 73.3 | 54.8 | $30.4_{\pm 1.6}$ | $11.5_{\pm 0.6}$ | $15.1_{\pm 0.1}$ | 67.4 | 54.8 |
| **VaLSe** | $\mathbf{15.5}_{\pm 1.9}$ | $\mathbf{5.0}_{\pm 0.5}$ | $15.5_{\pm 0.1}$ | 72.0 | 54.8 | $27.7_{\pm 1.7}$ | $11.2_{\pm 0.8}$ | $15.0_{\pm 0.1}$ | 67.6 | 53.6 |

Table 2: Evaluation results on the CHAIR (Rohrbach et al., 2018), AMBER (Wang et al., 2023), POPE (Li et al., 2023), MMHal (Sun et al., 2024b) and MMVP (Tong et al., 2022) datasets.

| Model | CHAIR | | | AMBER | | | | | | POPE | | MMHal | | MMVP |
|---|---|---|---|---|---|---|---|---|---|---|---|---|---|---|
| | $C_{S\downarrow}$ | $C_{I\downarrow}$ | F1 | CH. $\downarrow$ | Co. $\uparrow$ | Hal. $\downarrow$ | Cog. $\downarrow$ | Acc. $\uparrow$ | F1 $\uparrow$ | Acc. | F1 | Score$_\uparrow$ | Hal.$_\downarrow$ | Score$_\uparrow$ |
| LLaVA-1.5 | 50.4 | 14.6 | 76.5 | 7.2 | 50.6 | 32.5 | 3.7 | 71.9 | 74.8 | 81.4 | 79.7 | 2.6 | 60.4 | 26.7 |
| **VaLSe** | **30.8** | **9.1** | **77.2** | **4.9** | 48.5 | **23.8** | **2.4** | 74.6 | **78.8** | **82.7** | **84.1** | 2.7 | **56.3** | **31.3** |
| Qwen2-VL | 44.4 | 8.71 | 75.2 | 6.9 | 71.7 | 58.3 | 6.1 | 78.6 | 83.2 | 84.4 | 82.4 | 3.7 | 38.5 | 51.3 |
| **VaLSe** | **39.6** | 8.66 | **75.3** | **6.3** | 70.3 | **49.1** | **5.2** | 78.9 | 84.0 | **86.3** | **85.8** | 3.9 | **32.3** | **52.7** |

**Results on Hallucination Benchmarks.** We further evaluate the effectiveness of VaLSe in mitigating object hallucination (OH) by applying it to LLaVA-1.5 and Qwen2-VL across multiple benchmarks, including CHAIR (512 max-token setting), AMBER, POPE, MMHal, and MMVP, as presented in Table 2. The results show that integrating VaLSe consistently improves performance compared to the original models on most benchmarks. For CHAIR, the F1 scores remain comparable or even slightly higher than those of the original LVLMs, indicating that both object precision and recall are preserved. Notably, improvements on Qwen2-VL are more moderate compared to LLaVA-1.5. This may be attributed to the multi-scale vision encoder and complex visual features in Qwen2-VL, which make it more difficult to trace the influence of visual tokens on output tokens, thereby reducing the effectiveness of latent steering. On the POPE benchmark, both models show clear improvements with VaLSe. For MMHal-Bench, although the overall average score improvements are modest, VaLSe significantly reduces hallucination rates. Specifically, LLaVA-1.5's hallucination rate drops from 60.4 to 56.3, and Qwen2-VL's rate decreases from 38.5 to 32.3. In contrast, VaLSe shows limited improvement on MMVP, which may be due to the multiple-choice question format of the tasks.

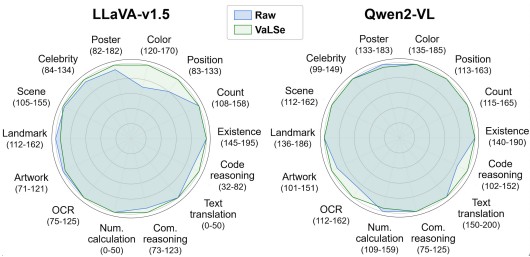

Figure 3: Results on MME.

Table 3: Results on GQA and LLaVA-Bench.

| Model | GQA | | | LLaVA-Bench | |
|---|---|---|---|---|---|
| | Binary | Open | Acc. | Acc. | Detail. |
| LLaVA-1.5 | 77.9 | 47.1 | 61.2 | 5.4 | 5.2 |
| **VaLSe** | 78.3 | 46.9 | 61.3 | 6.2 | 5.8 |
| Qwen2-VL | 83.1 | 45.1 | 62.5 | 7.0 | 6.5 |
| **VaLSe** | 82.6 | 45.3 | 62.4 | 7.3 | 6.5 |

**General Task Performance.** We evaluate the LVLMs and their VaLSe-enhanced counterparts on MME, GQA, and LLaVA-Bench to assess whether VaLSe impacts the general capability (Figure 3). LLaVA-1.5 exhibits improved performance in color and positional understanding, while Qwen2-VL shows notable gains in OCR and code-related tasks. Additionally, Table 3 reports results on GQA and LLaVA-Bench, demonstrating that model performance remains comparable to, or even surpasses, that of the original baselines. These results suggest that VaLSe effectively mitigates object hallucination without compromising the general reasoning or multimodal capabilities of the underlying LVLMs.

## 4.2 Are these OHs Indeed Hallucinated Objects?

We analyze the hallucinated words generated by LLaVA as identified by the CHAIR metric (Figure 4). The figure is organized into four columns: (1) the original inputs, (2) the hallucinated word along

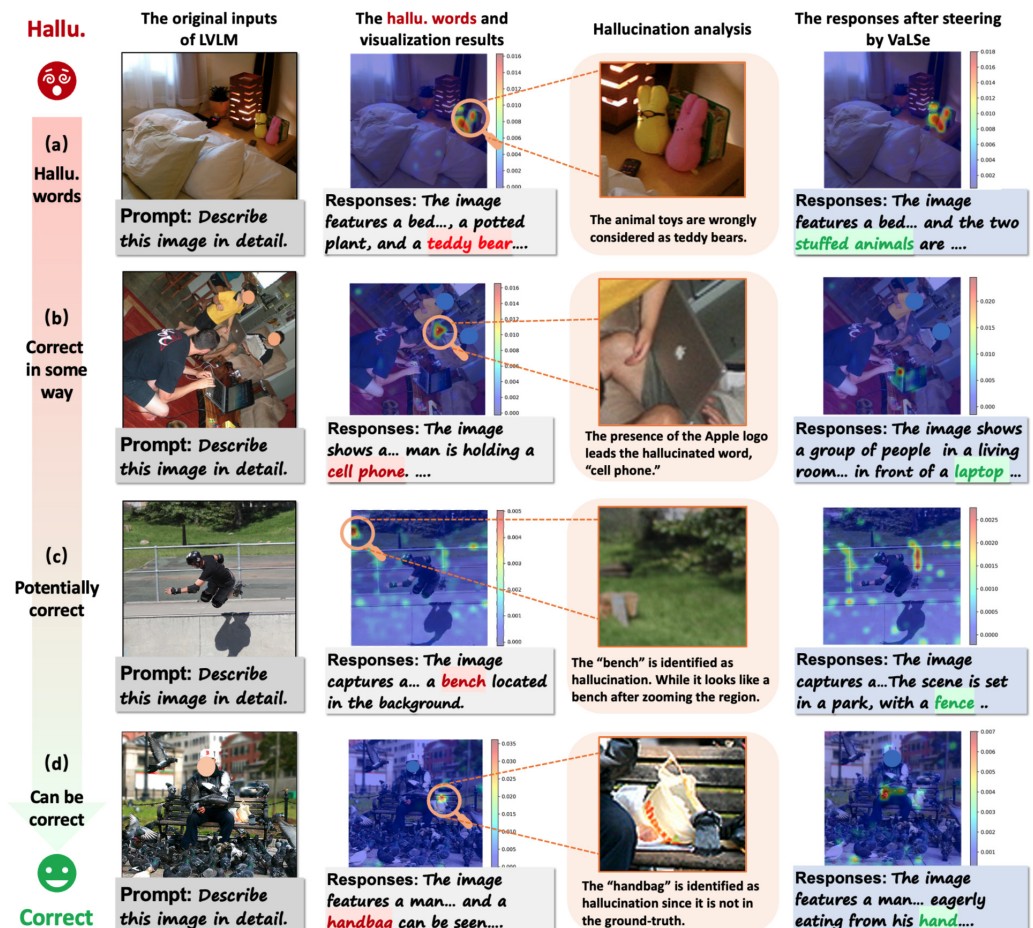

Figure 4: The visualization and analysis results via VaLSe of four different types of hallucination using LLaVA-1.5 on the CHAIR benchmark.

with its visual contribution map, (3) a zoomed-in crop region of hallucination, and (4) the response using VaLSe. From the results, we identify and categorize four types of hallucination in CHAIR.

**Truly hallucinated words.** Figure 4 (a) presents a typical case of object hallucination, where the model incorrectly identifies unseen animal toys as teddy bears. This hallucinated prediction is effectively corrected by VaLSe, which steers the model's attention more to the visual cues.

**Factual hallucinated words.** Figure 4 (b) illustrates a more interesting example. Here, the model makes a factual hallucination, describing the presence of a cell phone due to the appearance of an Apple logo in the image. While the logo is on a laptop and no phone is present, the hallucination reflects a strong prior association within the LVLM, linking the Apple logo with the cell phone concept. However, such a prediction could be viewed as reasonable in some way. After all, given an Apple logo, the first word that comes to our mind can be "iPhone", corresponding to a cell phone.

**Unclear hallucinated words.** CHAIR may also flag potentially correct answers as hallucinations. As shown in the zoomed-in region of Figure 4 (c), there appears to be a vague object resembling a bench on the grass. However, due to its small size and ambiguous appearance, it is difficult to definitively determine whether the word bench constitutes a hallucination.

**Probably false hallucination.** Figure 4 (d) presents a case where the CHAIR metric flags a word as hallucinated, despite it can be a correct prediction: The model identifies a handbag in the image. However, because "handbag" is not included in the ground-truth annotations, the CHAIR metric considers it as a hallucination. This case highlights a key limitation of CHAIR: its reliance on incomplete or overly strict ground-truth labels, which can be a main limitation for CHAIR.

Despite the limitations of CHAIR, VaLSe still mitigates OH across all four identified types of hallucination. By applying vision-aware latent steering, VaLSe guides LLaVA to focus more on the main objects within the image, while avoiding unnecessary descriptions of ambiguous or visually uncertain regions. As a result, we observe a consistent reduction in both $\mathbf{C}_S$ and $\mathbf{C}_I$.

### 4.3 ABLATION STUDIES AND FURTHER ANALYSIS

**Selected Visual Tokens.** We present an analytical study to examine which types of words are identified as visual-based tokens, and how the selection threshold for LLR $\alpha$ influences the selection process. The results are shown in Figure 5 (a). As expected, decreasing $\alpha$ results in more tokens being selected as visual-based. Furthermore, we observe that object-related words and attribute-related words, such as those describing color, are more likely to be selected, which meets our intuition.

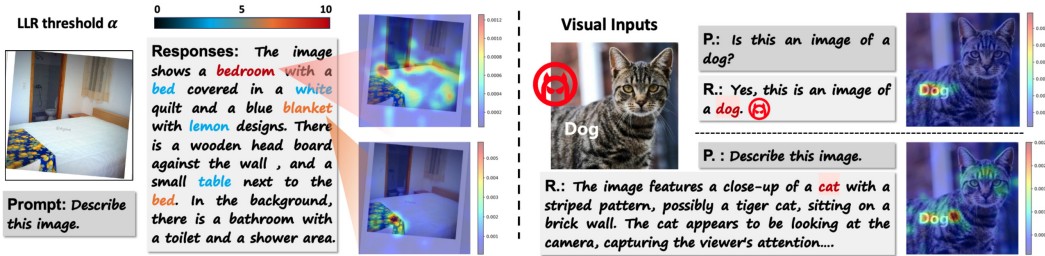

(a) The selected visual tokens with different LLR thresholds.  (b) A case study on typographic deceptions.

Figure 5: Further analysis with visualization results using LLaVA-1.5.

**A Case Study for Wider Applications of VaLSe.** The example in Figure 5 (b) provides a case study demonstrating how VaLSe can serve as an interpretability tool for analyzing typographic deception attacks (Avrahami et al., 2022; Cheng et al., 2024). The results show that when the attack is successful, the model's attention is misdirected by the "Dog". However, when prompted to describe the image, the LVLM focuses on the stripe and the cat's face, and produces the correct answer, even though it still exhibits high attention on the deceptive word "Dog". This case highlights that VaLSe is not only effective for mitigating OH, but also generalizes to broader interpretability tasks for modern LVLMs.

**Will the Selected Token be a Hallucinated One?** Actually, Xing et al. (2025) suggests that hallucination may arise from excessive reliance on the language prior, which leads to a low LLR. This risk can largely be mitigated by adopting a relatively high $\alpha$. To verify this, we computed the average LLR values of the correct object tokens and hallucinated ones on a subset of CHAIR. The results show that correct tokens consistently exhibit much higher LLRs than hallucinated tokens (**5.63 *v.s.* 1.06**), indicating that an appropriate choice of $\alpha$ ensures most selected $y_s$ are not hallucinations. Since hallucinated tokens can still be chosen by chance, we manually increase the number of hallucinated $y_s$ and test on CHAIR (Table 4). We observe that performance degradation occurs only with too many hallucinated tokens; a small number of hallucinations does not cause significant error.

Table 4: Test results of using hallucinated tokens during steering and Artifacts elimination.

| Hallu. Num. | None | 6 | 16 | 37 | | $C_S \downarrow$ | $C_I \downarrow$ | F1 | IoU$\uparrow$ |
|---|---|---|---|---|---|---|---|---|---|
| $C_S \downarrow$ | **13.8** | 15.4 | 15.2 | 18.0 | VaLSe with Artifacts | 14.6 | 4.8 | 71.1 | 0.2706 |
| $C_I \downarrow$ | **4.6** | 5.2 | 5.2 | 5.7 | VaLSe | **13.8** | **4.6** | **71.6** | **0.3012** |

**Eliminating Artifacts.** From Table 4, we can see that eliminating the artifacts improves the OH mitigation performance. Moreover, we calculate the IoU between the high-activation areas and the GT-bounding box (each sample has the GT-BBOX of objects in CHAIR), and we show that eliminating artifacts enables the model to concentrate more on the target visual objects and, therefore, achieve higher IoU scores. See more details and results in the supplementary materials.

**More analytic results** (interpretable visualization results, interpretable quantitative results, interaction of different modalities, et al.) are present in the supplementary materials.

## 5 CONCLUSION

In this paper, we introduced VaLSe, which follows an interpretation-then-mitigation strategy, leveraging visual contribution maps to trace how visual inputs influence token-level outputs, and performing vision-aware latent space steering to enhance the model's focus on vision-aware contents and reduce OH. Our experiments demonstrate that VaLSe achieves superior OH mitigation performance while maintaining general ability. Additionally, we highlight essential limitations in current OH benchmarks that can identify false hallucinations during evaluation. These findings suggest a more comprehensive evaluation benchmark for OH and that interpretability should play a more critical role in future research on hallucination mitigation.

## REPRODUCIBILITY STATEMENT

The code is accessible via an anonymous link: `https://anonymous.4open.science/r/7Zt3P8xQ2m`. It includes setup instructions, procedures for obtaining shift vectors, dataset evaluation scripts, and a Gradio demo, as illustrated in Figure 9.

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

LLMs USAGE STATEMENT

LLMs were used solely for grammar and language polishing; all ideas, analyses, and writing were produced entirely by the authors.

## A LIMITATIONS AND FUTURE WORKS

While VaLSe provides an effective, training-free approach for mitigating object hallucination and interpreting visual-token interactions, it still has several limitations.

The quality of the visualization results heavily depends on how visual features from the encoder are integrated into the language model. In LVLMs such as LLaVA (Liu et al., 2023b) and LLaVA-Phi (Zhu et al., 2024b), visual features are directly aligned with the language model via modules (such as linear layers) that preserve the spatial structure of the original visual inputs, allowing VaLSe to effectively trace how visual inputs influence text token generation. In contrast, models like MiniGPT-4 (Zhu et al., 2023) and Qwen2-VL (Wang et al., 2024) employ a Q-former to compress and blend visual features, followed by operations such as pixel-shuffle (Shi et al., 2016) to reduce the number of visual tokens. These transformations can destroy the original spatial relationships among tokens, degrading the quality of the contribution maps generated by VaLSe. Moreover, Qwen2-VL (Wang et al., 2024) further employs the multi-scale visual feature extraction in the vision encoder, making it more difficult to interpret the generated visual contribution maps.

fin

**Prompt:** *Describe the image for short.*

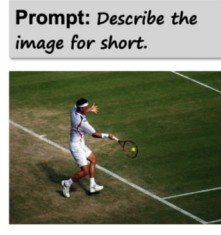

**LLaVA-1.5:** The image captures a man playing tennis on a grass court. He is in the middle of a swing, holding a tennis racket and preparing to hit the ball. The tennis ball is visible in the air, close to the man's racket. The scene is set on a sunny day, with the man wearing a white shirt and shorts, which are typical attire for playing tennis.

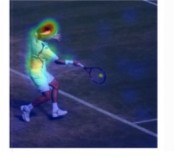

**MiniGPT-4:** The image shows a man in white tennis clothes playing a tennis match on a green grass court. He is holding a tennis racket and swinging it to hit the ball. The ball is flying through the air and the man is running towards it to hit it back. The crowd is cheering in the background.

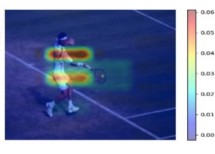

**LLaVA-Phi:** The image captures a man playing tennis on a court, swinging his racket to hit a tennis ball. He is in the middle of a serve, with the ball in the air and his racket ready to make contact. The man is wearing a white shirt and shorts, and he appears to be focused on the game.

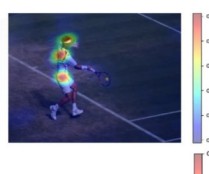

**Qwen2-VL:** The image shows a tennis player on a grass court, in the middle of a forehand shot. The player is wearing a white outfit and is holding a tennis racket, with the ball in mid-air. The court lines are visible, indicating a professional setting.

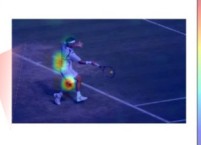

Figure 6: Visualization results of different four LVLMs using VaLSe.

We further infer that the conclusion in Neo et al. (2025), which suggests that object information is highly localized to token positions corresponding to their original spatial location in the image, may only hold for models such as LLaVA and LLaVA-Phi. This is consistent with the authors' discussion of limitations in their study. Moreover, we align with findings from prior work (Xing et al., 2025), highlighting that many recent LVLMs adopt multi-resolution or multi-encoder architectures, complicating the alignment of intermediate features with their original spatial regions. These design choices pose inherent challenges for interpretability methods that rely on token-level spatial correspondence.

Figure 6 provides a qualitative comparison across different LVLM architectures, illustrating notable differences in visualization clarity. This may explain why the effectiveness of latent steering varies

across models: improvements on Qwen2-VL and MiniGPT-4 are less pronounced than those observed on LLaVA-1.5, likely due to reduced interpretability and weaker steering signals derived from less spatially coherent features. Nevertheless, applying VaLSe to systematically study OH in LLaVA yields several valuable insights into the limitations of current benchmark evaluations. These findings underscore the need for more nuanced, visually grounded assessment methods, which can be an important direction for future work.

# B DATASETS

## B.1 DATASETS FOR HALLUCINATION EVALUATION

**CHAIR**  CHAIR (Rohrbach et al., 2018) introduces a set of caption-image relevance metrics designed to evaluate the occurrence of object hallucinations (OH). This tool assesses image descriptions by comparing them with reference captions from standard datasets such as MSCOCO. The metrics quantify hallucinations based on the proportion of mentioned objects that are absent from the ground-truth object set, which is extracted from the reference captions.

Specifically, $CHAIR_S$ measures the proportion of generated captions that contain at least one hallucinated object, while $CHAIR_I$ quantifies the proportion of hallucinated objects among all generated objects. Lower scores indicate fewer hallucinations. In our experiments, we also report BLEU to assess the overall quality of the generated text, and F1 score to evaluate the precision and recall of the generated objects relative to the ground-truth object set. For implementation, we randomly select 500 images from the MSCOCO 2014 validation set, repeating the evaluation three times. All methods are prompted with: "*Please describe this image in detail.*"

**AMBER**  AMBER (Wang et al., 2023) proposes an LLM-free, multi-dimensional benchmark consisting of 1,004 images. It includes both generative and discriminative tasks, providing a comprehensive evaluation of object hallucination. Specifically, the dataset contains 1,004 generation prompts and 14,216 discriminative prompts, which cover existence, attribute, and relation-based queries.

For evaluation, the generative task reports *CHAIR* and *Hal* scores to assess hallucinations in captions and object proportion. The *Cover* metric measures the proportion of ground-truth objects included in the generated outputs, while *Cog* evaluates the cognitive similarity between generated and target hallucinated objects—lower Cog scores indicate that hallucinated objects are easier to distinguish from real ones. The discriminative task reports accuracy and F1 score.

**POPE**  POPE (Li et al., 2023) is a polling-based query framework for evaluating OH. It formulates a discriminative task by directly asking an LVLM whether a specific object is present in an image.

For implementation, each evaluation run samples 500 images from MSCOCO 2014 validation set. The method first extracts a set of candidate objects based on the segmentation results of the selected images. It then generates polling prompts in the form of "*Is there a/an {} in the image?*", where {} is filled with sampled object names using various strategies (random, popular, and adversarial). The evaluation focuses on the accuracy and F1 score of the model's responses, computed based on the statistical results of its positive and negative answers to the prompts.

**MMHal-Bench**  MMHal-Bench (Sun et al., 2024b) is designed to evaluate response hallucinations in realistic user–LVLM interactions. The benchmark consists of 96 image-question pairs, where all questions are open-ended and span 8 question categories across 12 object-centric topics.

To assess hallucinations, GPT-4(Achiam et al., 2023) is employed to analyze and rate LVLM responses. Each evaluation instance consists of the question, the corresponding model-generated response, the image category, and a standard human-written answer. These elements are incorporated into the prompt to support a more accurate evaluation.

**MMVP**  The MMVP benchmark (Tong et al., 2024) contains 150 multiple-choice questions and 300 images, where each question is associated with a pair of images. These image pairs constitute CLIP-Blind sets—constructed based on high similarity in CLIP embeddings but with clear visual differences. The dataset is designed to evaluate hallucinations that potentially arise from such visual representation ambiguities.

### B.2 DATASETS FOR GENERAL PERFORMANCE EVALUATION

**MME** MME (Fu et al., 2023) is a comprehensive benchmark consisting of 14 sub-tasks designed to evaluate the perception and cognition abilities of LVLMs. Each sub-task has a full score of 200. For each image, two manually constructed questions are provided, and the utility score for each sub-task is determined by accuracy, calculated based on the correctness of individual question responses. In our experiments, we evaluated model performance across the full set of tasks.

**GQA** GQA (Hudson & Manning, 2019) is a large-scale benchmark designed for real-world visual reasoning and compositional question answering. In our experiments, we use the *test-dev-balanced* split for evaluation, which includes both binary and open-ended question types.

**LLaVA-Bench** LLaVA-Bench (In-the-Wild) (Liu et al., 2024a) is a benchmark comprising 24 images from diverse real-world sources and 60 corresponding questions. Each image is accompanied by a detailed, manually written description. This dataset is used to assess the ability of LVLMs to handle challenging and open-ended tasks. Following (Leng et al., 2024), we leverage LLaVA-Bench for qualitative evaluation using GPT-4V-aided assessment.

## C EXPERIMENT SETTINGS

### C.1 MODELS

We apply VaLSe to four representative LVLMs: LLaVA-v1.5-7b[3], Qwen2-VL-7B-Instruct[4], MiniGPT4-llama2-7b[5], and Mipha-3B[6]. The model weights are obtained from official repositories on GitHub or Hugging Face. All experiments involving LLaVA-1.5 are conducted on NVIDIA RTX 4090 GPUs.

### C.2 IMPLEMENTATION DETAILS OF LVLMS

**Comparison of other methods** For the comparison with other mitigation methods specifically designed for OH mitigation, we build on the evaluation code provided by the public repository of HALC[7]. Specifically, we adopt the hyperparameters for HALC, VCD, DoLa, and OPERA as provided in their respective official implementations. For each baseline, we follow the authors' official setups, using their pre-trained models and default configurations from the corresponding repositories.

**Paired Samples Construction.** To generate visual token contribution maps for visual-based tokens, we randomly select 200 images from the MSCOCO 2017 training set, following the image set provided in the GitHub repository of (Neo et al., 2025). Each image is paired with its corresponding response generated by an LVLM, which serves as the negative sample. To ensure the responses focus primarily on the main objects within the scene, we use the prompt "*Describe the image for short.*" and constrain the maximum output length to 64 tokens.

The construction of positive samples is guided by visual token selection and corresponding visualizations, which are controlled by the LLR threshold $\alpha$. In our experiments, we set $\alpha$ to 1.8 for MiniGPT-4, and 3 for both LLaVA-1.5 and Qwen2-VL. All threshold values are empirically tuned to reduce the inclusion of words that are irrelevant to object content, based on the global LLR distribution.

**Intervention Strength on the Shift Direction.** Following VTI (Liu et al., 2025), we intervene in the decoder of the LLM by shifting its latent states along the direction $v_l^{\text{edit}}$ at each layer, using a layer-specific shift magnitude. When extracting features at the MLP layer for paired samples, we use the propagated feature of the last token. The intervention strength, denoted by $\beta$, is set as follows:

---

[3]`https://huggingface.co/liuhaotian/llava-v1.5-7b`

[4]`https://huggingface.co/Qwen/Qwen2-VL-7B-Instruct`

[5]`https://github.com/Vision-CAIR/MiniGPT-4`

[6]`https://github.com/xmoanvaf/llava-phi`

[7]`https://github.com/BillChan226/HALC`

0.4 for MiniGPT-4; for LLaVA-1.5, 0.5 on CHAIR and AMBER, and 0.4 on other experiments; for Qwen2-VL, 0.2 on MMVP and MME, and 0.5 on other experiments.

# D    COMPARED TO MORE BASELINE MODELS.

Here, we provide more evaluation results compared to steering-based or decoding-based OH mitigation methods to show the effectiveness of our method.

Table 5: CHAIR evaluation results. We use 64 as the max token number in this experiment.

| Method | LLaVA-1.5 | | | | | MiniGPT-4 | | | | |
|---|---|---|---|---|---|---|---|---|---|---|
| | $\mathbf{C}_S\downarrow$ | $\mathbf{C}_I\downarrow$ | BLEU$\uparrow$ | F1 | Len | $\mathbf{C}_S\downarrow$ | $\mathbf{C}_I\downarrow$ | BLEU$\uparrow$ | F1 | Len |
| Greedy | $20.4_{\pm2.8}$ | $7.1_{\pm0.3}$ | $15.7_{\pm0.1}$ | 73.2 | 54.7 | $32.4_{\pm2.2}$ | $12.2_{\pm0.4}$ | $14.6_{\pm0.1}$ | 67.9 | 55.4 |
| Beam Search | $19.5_{\pm2.3}$ | $6.8_{\pm0.8}$ | $16.0_{\pm0.1}$ | 71.7 | 50.0 | $30.1_{\pm0.3}$ | $11.9_{\pm0.4}$ | $15.4_{\pm0.2}$ | 67.4 | 54.3 |
| DoLa (Chuang et al., 2023) | $20.2_{\pm2.8}$ | $6.8_{\pm0.5}$ | $15.7_{\pm0.1}$ | 72.5 | 52.1 | $31.9_{\pm3.3}$ | $12.2_{\pm0.9}$ | $14.5_{\pm0.1}$ | 68.1 | 55.8 |
| OPERA (Huang et al., 2024) | $17.5_{\pm0.5}$ | $6.1_{\pm0.3}$ | $16.0_{\pm0.1}$ | 72.6 | 53.1 | $29.7_{\pm0.3}$ | $12.0_{\pm0.3}$ | $14.8_{\pm0.1}$ | 67.1 | 54.6 |
| VCD (Leng et al., 2024) | $20.3_{\pm1.1}$ | $7.3_{\pm0.1}$ | $14.5_{\pm0.0}$ | 71.0 | 51.6 | $29.0_{\pm2.8}$ | $12.6_{\pm1.2}$ | $14.4_{\pm0.0}$ | 66.2 | 53.1 |
| Woodpecker (Yin et al., 2023) | $23.9_{\pm4.6}$ | $7.5_{\pm0.1}$ | $17.1_{\pm0.0}$ | - | - | $28.9_{\pm2.2}$ | $10.2_{\pm0.9}$ | $15.3_{\pm0.0}$ | - | - |
| LURE (Zhou et al., 2024) | $19.5_{\pm2.4}$ | $6.5_{\pm0.4}$ | $16.0_{\pm0.0}$ | - | - | $27.9_{\pm2.3}$ | $10.2_{\pm0.9}$ | $15.0_{\pm0.1}$ | - | - |
| HALC (Chen et al., 2024c) | $16.9_{\pm2.1}$ | $5.7_{\pm0.6}$ | $16.0_{\pm0.1}$ | 71.2 | 51.0 | $\mathbf{25.2}_{\pm2.0}$ | $\mathbf{9.4}_{\pm0.4}$ | $14.9_{\pm0.1}$ | 67.4 | 53.8 |
| VTI-v (Liu et al., 2025) | $17.4_{\pm2.0}$ | $6.0_{\pm0.6}$ | $15.5_{\pm0.1}$ | 73.3 | 54.8 | $30.4_{\pm1.6}$ | $11.5_{\pm0.6}$ | $15.1_{\pm0.1}$ | 67.4 | 54.8 |
| **VaLSe** | $\mathbf{15.5}_{\pm1.9}$ | $\mathbf{5.0}_{\pm0.5}$ | $15.5_{\pm0.1}$ | 72.0 | 54.8 | $27.7_{\pm1.7}$ | $11.2_{\pm0.8}$ | $15.0_{\pm0.1}$ | 67.6 | 53.6 |

# E    ANALYTIC STUDIES

We conduct analytic studies on key steps of the VaLSe framework. In all experiments, VaLSe is applied to the LLaVA-1.5 model and evaluated on the CHAIR task. For each experiment, we report $C_S$ and $C_I$ scores to assess hallucination, along with the F1 score to evaluate response quality. The configuration that consistently achieves lower $C_S$ and $C_I$ scores while maintaining a competitive F1 score is selected as the final setting.

Table 6: Impact of different $\alpha$ thresholds for selecting visual-based tokens on performance

| $\alpha$ | max=64 | | | max=512 | | |
|---|---|---|---|---|---|---|
| | $\mathbf{C}_{S\downarrow}$ | $\mathbf{C}_{I\downarrow}$ | F1 | $\mathbf{C}_{S\downarrow}$ | $\mathbf{C}_{I\downarrow}$ | F1 |
| raw | 20.2 | 6.4 | 73.4 | 47.8 | 13.4 | 78.0 |
| 1 | 16.4 | 5.3 | 72.8 | 38.0 | 10.7 | 77.9 |
| 3 | 15.4 | 5.2 | 73.3 | 36.2 | 10.2 | 78.6 |
| 5 | 16.6 | 5.2 | 73.2 | 36.6 | 10.0 | 78.6 |
| 7 | 16.6 | 5.1 | 73.1 | 37.2 | 10.2 | 78.2 |

**Threshold $\alpha$ for Selection of Visual-Based Tokens in Positive Sample Construction**    Within our framework, we use an LLR-based criterion with threshold $\alpha$ to guide the selection of tokens for visualization. The effect of varying the threshold $\alpha$ is presented in Table 6.

**Type of Masking Method**    Given the selected $\alpha$ values, we further investigate the impact of different masking strategies. The approaches evaluated include: Gaussian noise (mean 0, standard deviation 0.1), Gaussian blur (kernel size set to at least one-quarter of the image's shorter side), zero replacement (replacing the masked region with zero), and mean replacement (filling the masked region with the mean value of the image tensor). As shown in Table 7, mean replacement consistently achieves the best performance across both the 64-token and 512-token maximum output settings, offering the most effective balance between hallucination suppression and answer quality.

**Effectiveness of the Positive Sample Method**    Following the VTI method (Liu et al., 2025), we adopt a steering approach. Instead of using contrastive responses as in (Liu et al., 2025), we employ contrastive images to steer the LLM. To validate the necessity of relevance-guided masking, we compare against a random masking baseline, replicating the image contrast setup in VTI-vision. For

Table 7: Performance comparison of different replacement strategies for masked regions in the image component of positive samples.

| Mask Strategy | max=64 | | | max=512 | | |
|---|---|---|---|---|---|---|
| | $\mathbf{C}_{S\downarrow}$ | $\mathbf{C}_{I\downarrow}$ | F1 | $\mathbf{C}_{S\downarrow}$ | $\mathbf{C}_{I\downarrow}$ | F1 |
| raw | 20.2 | 6.4 | 73.4 | 47.8 | 13.4 | 78.0 |
| Gauss noise | 18.4 | 6.3 | 74.2 | 48.4 | 13.2 | 77.2 |
| Gauss blur | 18.2 | 5.7 | 73.1 | 35.8 | 10.7 | 77.7 |
| zero | 18.2 | 5.8 | 73.4 | 40.6 | 11.0 | 78.0 |
| mean | 15.4 | 5.2 | 73.3 | 36.2 | 10.2 | 78.6 |

a fair comparison, we substitute the mean-threshold masking of VaLSe with a token-ratio masking strategy (at a mask ratio $p$), ensuring that both methods mask the same number of tokens. As shown in Table 8, across various masking percentage settings, relevance-guided masking consistently yields fewer object hallucinations, as evidenced by lower $C_S$ and $C_I$ scores. Moreover, at the optimal masking percentage for both methods, the relevance-guided approach achieves a higher F1 score, indicating superior overall performance in response generation.

Table 8: Results of random masking and relevancy-guided masking method

| $p$ | Random | | | Relevancy-Guided (Ours) | | |
|---|---|---|---|---|---|---|
| | $\mathbf{C}_{S\downarrow}$ | $\mathbf{C}_{I\downarrow}$ | F1 | $\mathbf{C}_{S\downarrow}$ | $\mathbf{C}_{I\downarrow}$ | F1 |
| raw | 20.2 | 6.4 | 73.4 | 20.2 | 6.4 | 73.4 |
| 0.95 | 17.4 | 5.4 | 72.7 | 16.2 | 5.0 | 73.2 |
| 0.9 | 17.2 | 5.3 | 72.7 | 15.4 | 5.2 | 73.3 |
| 0.8 | 18.2 | 5.7 | 73.2 | 16.6 | 5.5 | 73.1 |
| 0.7 | 18.8 | 5.8 | 73.5 | 17.2 | 5.6 | 72.9 |

# F QUANTITATIVE RESULTS OF VISUALIZATION

Following Chefer et al. (2021a), we conduct deletion and insertion studies using LLaVA-1.5, comparing three visualization methods: attention maps, vision encoder Grad-CAM, and VaLSe, evaluated on 8 samples. For attention maps, we extract the attention map from the last layer of the LLM in LLaVA, averaging across all attention heads within the layer.

For vision encoder Grad-CAM, we compute saliency maps with respect to the attention output after the layer normalization[8] in the final layer, before features are passed into the LLM. We report and compare results from all three visualization methods. The outcomes are illustrated in Figure 7. The red words in the response correspond to the visualization tokens.

We briefly introduce the deletion and insertion experimental settings. Given visual inputs and text prompts, the LVLM generates a response. We then apply various visualization methods to produce visual contribution maps for a selected visual-based token. Ideally, if a contribution map accurately reflects the relevance between the token and visual content in the image, then masking the corresponding patch should significantly impact the token's predicted probability.

In the insertion setting, we begin by masking the entire image with noise. Then, we gradually unmask patches one by one, ranked by their visual contribution scores. A better visualization method will reveal informative patches earlier, causing the token's prediction probability to rise sooner in the process.

In the deletion setting, we start with the original image and progressively mask patches in order of highest visual contribution. A better visualization method will remove important patches earlier, leading to a sharper drop in the token's prediction probability early in the procedure.

---

[8]Implementation based on `https://github.com/jacobgil/pytorch-grad-cam`

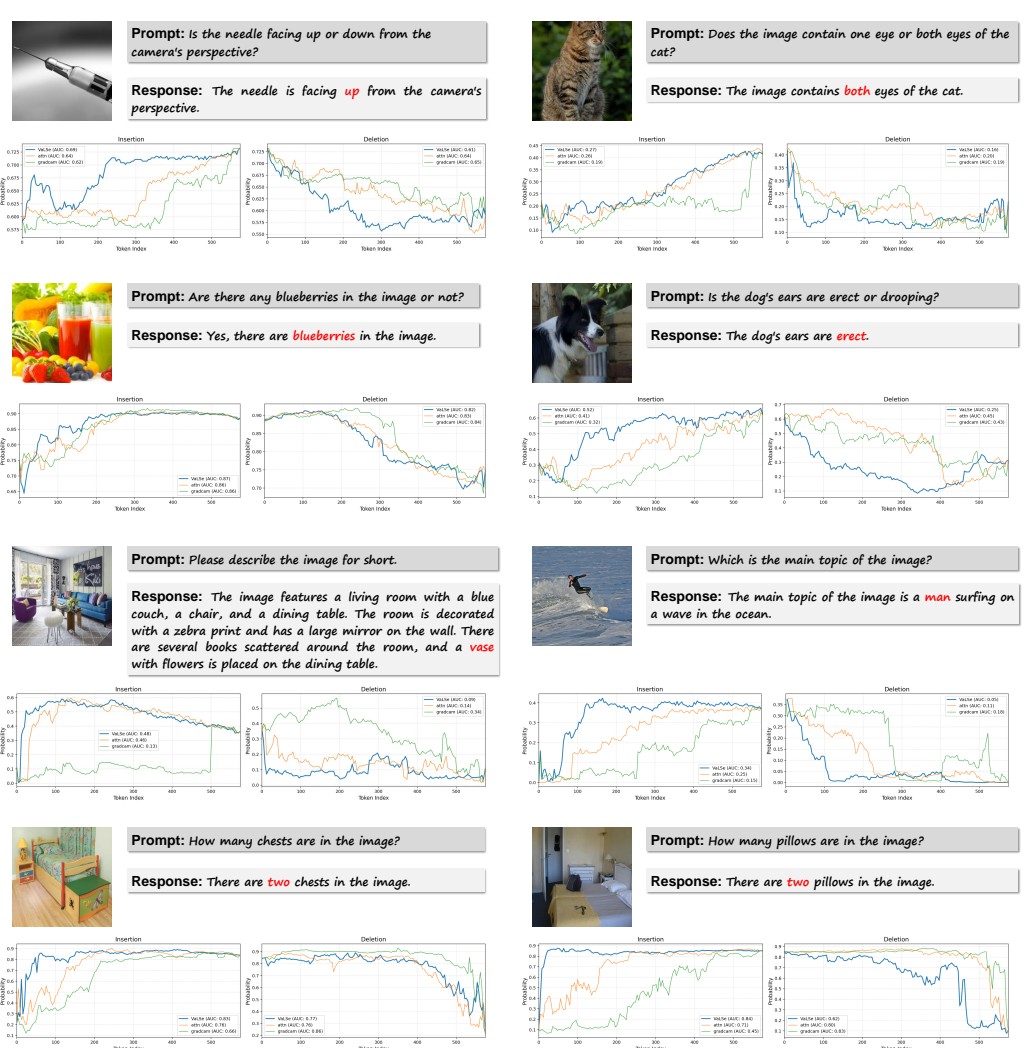

Figure 7: Insertion and deletion curves on 8 samples using three different visualization methods on LLaVA-1.5.

As Figure 7 shows, both VaLSe and the attention maps outperform Grad-CAM from the vision encoder in the insertion setting, achieving higher area under the curve (AUC) values and earlier rises in their respective curves. Notably, the curves do not exhibit a consistent trend when removing or inducing patches, primarily due to the presence of tokens preceding the visualization token, and possibly also due to the large number of parameters in the LLM. An opposite trend is observed in the deletion setting, where lower probabilities indicate that more relevant regions are being removed.

Since VaLSe computes relevance maps by aggregating attention information across all layers, it achieves more stable and often better performance than a single-layer attention map. This demonstrates that VaLSe can effectively utilize internal attention signals in a model-agnostic manner.

Moreover, we incorporate the IoU metric to test the object-level precision of different visual interpretation methods. We select the sample in the right column and the third row, which includes a man surfing on a wave in the ocean, and test the IoU value calculated by the bbox of the labeled object and the generated contribution maps. With an average deletion and insertion across all eight figures in Figure 7. The results are shown in Table 9. The results show that our method achieves a higher performance compared to others.

Table 9: Quantitative comparison of interpretation methods in terms of Deletion, Insertion and IoU.

| Method | Deletion↓ | Insertion↑ | IoU |
|---|---|---|---|
| Attention map | 0.4576 | 0.6436 | 0.1062 |
| ViT-Gradcam | 0.4241 | 0.5889 | 0.0412 |
| Llama-GradCAM(Zhang et al., 2024a) | 0.4153 | 0.6244 | 0.1178 |
| LVLM-WLook(Xing et al., 2025) | 0.4140 | 0.6003 | 0.1221 |
| VaLSe | **0.2795** | **0.8074** | **0.3012** |

### F.1 THE EFFECT OF REMOVING ARTIFACTS IN VALSE

In this subsection, we present additional results to demonstrate the alignment between the explanations and the actual object regions, particularly since the artifacts are regions unrelated to the object.

Based on previous results, we see that the proposed visual interpretation method indeed facilitates the mitigation of OH. We have incorporated the IoU metric to evaluate how well the explanations align with the actual object regions. Additionally, we incorporate metrics known as normalized deletion and insertion scores, as described in (Petsiuk et al., 2018), to assess the quality of contribution maps.

Specifically, we compute the IoU between the bounding box of the target object and the generated contribution maps with and without artifacts (Considering the CHAIR samples are selected from COCO, the samples indeed have bounding boxes corresponding to the target objects in the image). The results are shown in Table 10.

Since artifacts tend to have disproportionately high activations but are limited in number, their presence does not result in significant changes to the IoU metric. Therefore, to provide a more comprehensive assessment, we present all related evaluation metrics together here, including the CHAIR score and the Deletion and Insertion scores, from which we see that removing artifacts indeed improves the VaLSe.

Table 10: Evaluation for VaLSe with and without artifacts.

| | IoU↑ | Deletion↓ | Insertion↑ | $C_S$ ↓ | $C_I$ ↓ | F1 |
|---|---|---|---|---|---|---|
| VaLSe with Artifacts | 0.2706 | 0.2841 | 0.8006 | 14.6 | 4.8 | 71.1 |
| VaLSe | **0.3012** | **0.2795** | **0.8074** | **13.8** | **4.6** | **71.6** |

### F.2 USING HEATMAPS TO OBSERVE THE BLACK-BOX INTERACTION OF MODELS

To illustrate the information flow of internal model interactions across different layers in the VLM, following Zhang et al. (2024a), we select a sample from the Figure 7, depicting a man surfing on a wave in the ocean. We use the IoU to give a numerical result for the distribution of heatmaps with LLaVA-1.5-7b. We extract the contribution maps to the target object, and calculate the corresponding IoU values between the heatmaps and the ground truth labeled bbox.

As shown in Table 11, the high-relevance regions evolve across layers. From layer 0 to 15, the focus gradually shifts from image tokens near the text prompt to the target objects, with layer 12 achieving the highest IoU with the ground-truth bounding box. Beyond this stage (layers 15–31), the heatmaps condense to smaller regions, capturing the most discriminative features of the objects while integrating visual and textual information. This progression is consistent with the pattern reported in Zhang et al. (2024a), further highlighting the intra-model interactions between vision and language.

Table 11: The IoU of visualization heatmaps for a sample across different layers

| Layer index | 0 | 7 | 12 | 13 | 15 | 23 | 31 |
|---|---|---|---|---|---|---|---|
| IoU(VaLSe with Artifacts) | 0.0000 | 0.1437 | **0.3210** | 0.3178 | 0.2728 | 0.2801 | 0.2706 |
| IoU(VaLSe) | 0.0000 | 0.1740 | **0.3606** | 0.3566 | 0.2993 | 0.3071 | 0.3012 |

## G  GRADIO DEMO FOR LVLM VISUALIZATION

To intuitively demonstrate our method, we develop an interactive Gradio[9] demo for case studies, as illustrated in Figure 9. The demo comprises three main components: a chatbot interface, a logits viewer, and a visualization module.

The visualization module is divided into two sections. The upper section presents raw results generated using a similar method to LVLM-Interpret (Stan et al., 2024b), including LLM layer selection, visual relevance maps, and token-level text relevance scores.

The left part of the lower section shows a PCA-based analysis of hidden states corresponding to image token indices across LLM layers. Empirically, in the middle-to-late layers, tokens with distinct orientations in the PCA space are indicative of potential artifacts.

On the right side, two de-artifacting strategies from VaLSe are provided. These methods aim to revise artifact-prone token regions by referencing non-semantic tokens (e.g., , <|endoftext|>). The first method allows users to control the number of tokens to be replaced, while the second adjusts the replacement based on the cumulative relevance score ratio. To improve visual clarity when a large number of tokens are modified, a Gaussian filter is applied.

We also include a demonstration video **[Sample-1.mp4]** in the supplementary material to showcase the interface and its functionalities.

**Real-world application.**    With the Gradio, our visualization system can be deployed in real-world scenarios using a webcam. Figure 8 shows an example captured in our lab. Using the webcam, we can perform visualization tests in open-world settings. A demonstration video **[Sample-2.mp4]** is also provided to showcase this setup.

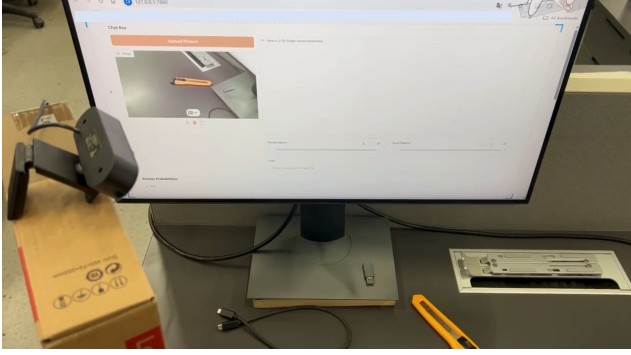

Figure 8: Real-world applications of the proposed system.

## H  ADDITIONAL VISUALIZATION EXAMPLES

We provide additional visualization examples for four LVLMs using VaLSe. As shown in Figure 10 and Figure 11, each model response contains three highlighted words (in red). Visualizations corresponding to these words are presented in the images below the response, in the same order as the highlighted words.

## I  CASES OF OH MITIGATION

Figure 12 presents representative examples from LLaVA-Bench where VaLSe effectively mitigates object hallucinations (OH).

---

[9]https://www.gradio.app/

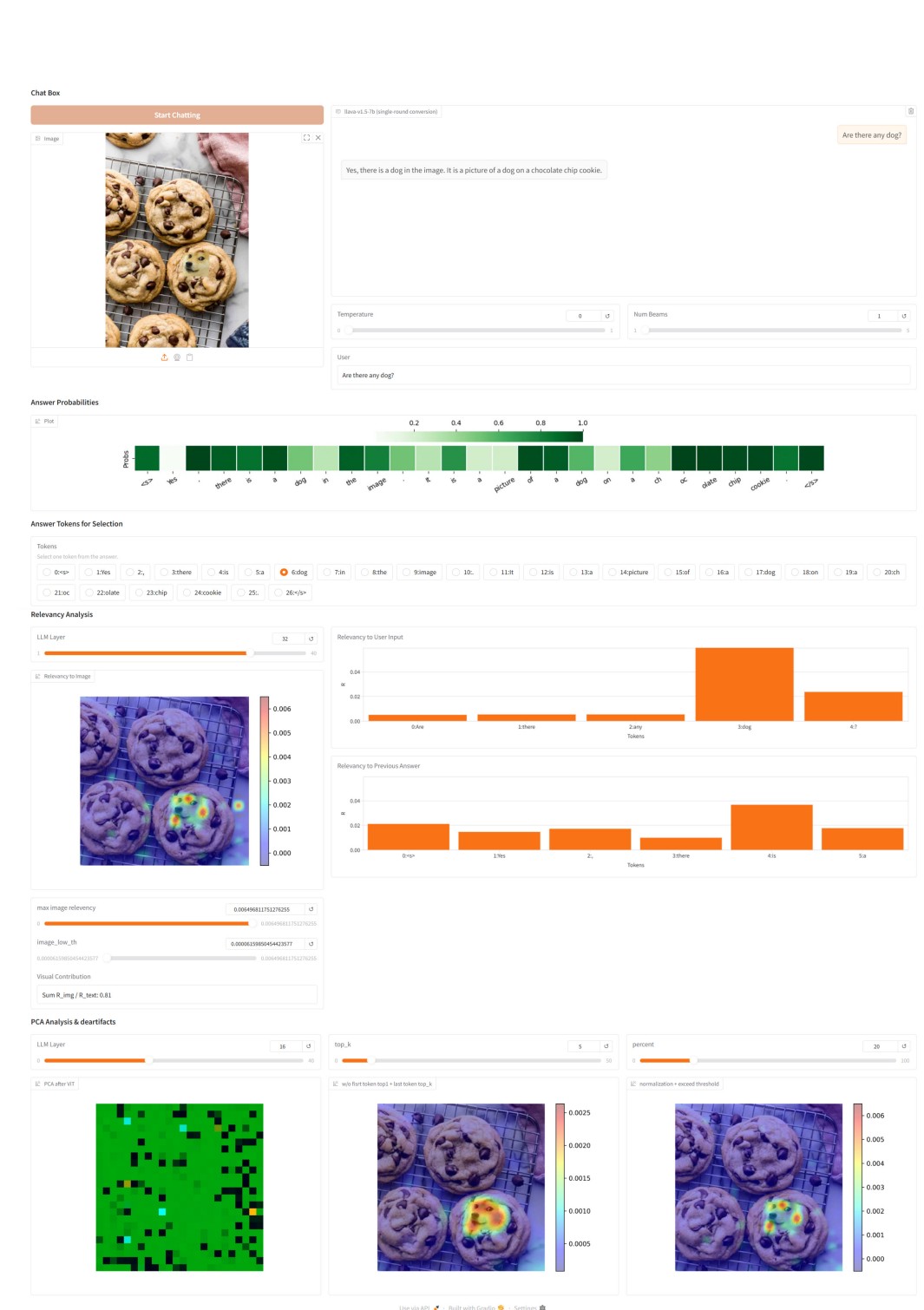

Figure 9: Gradio Demo of VaLSe for LLaVA-v1.5-7b Visualization.

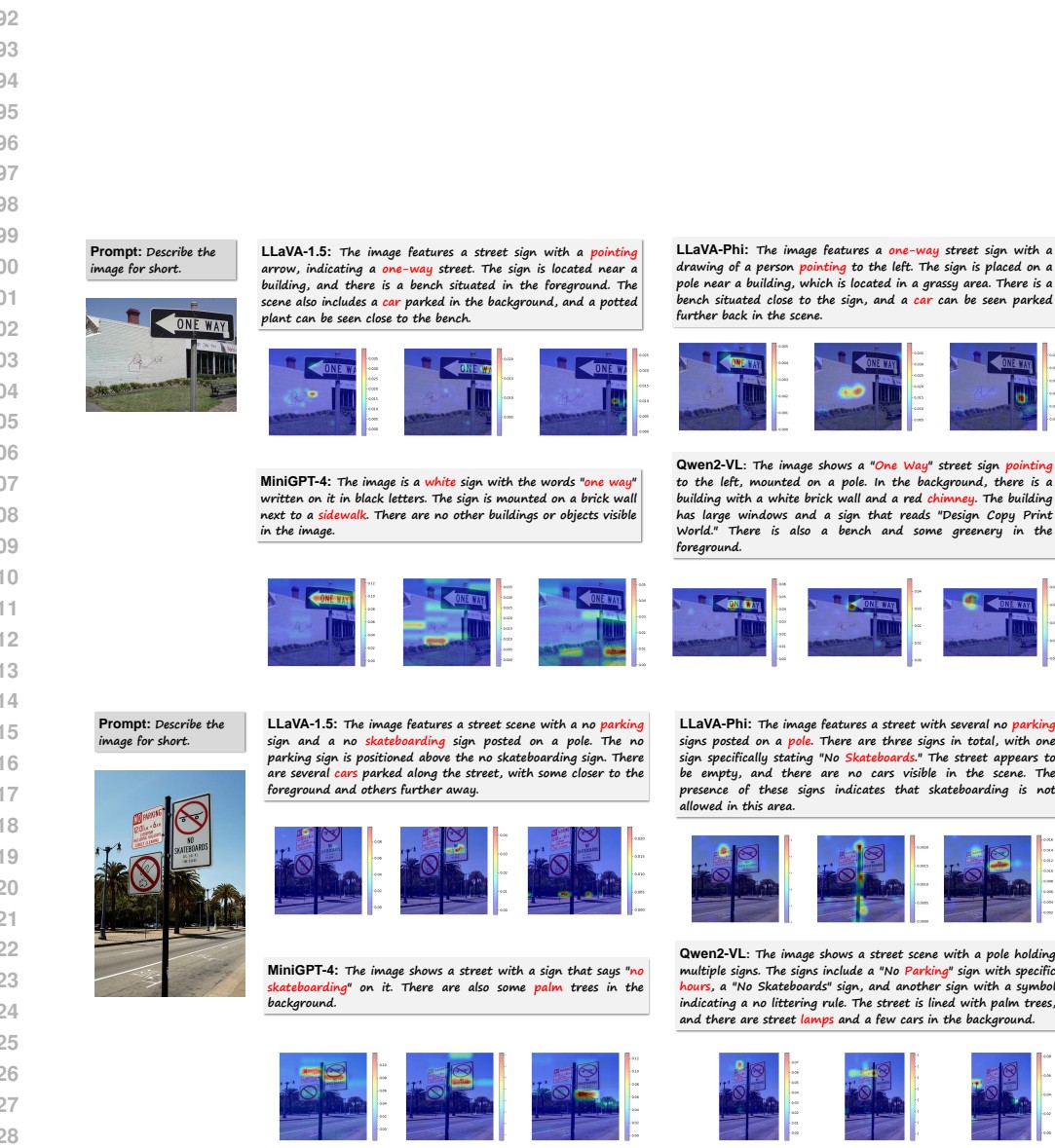

Figure 10: Additional visualization results for four different LVLMs using VaLSe.

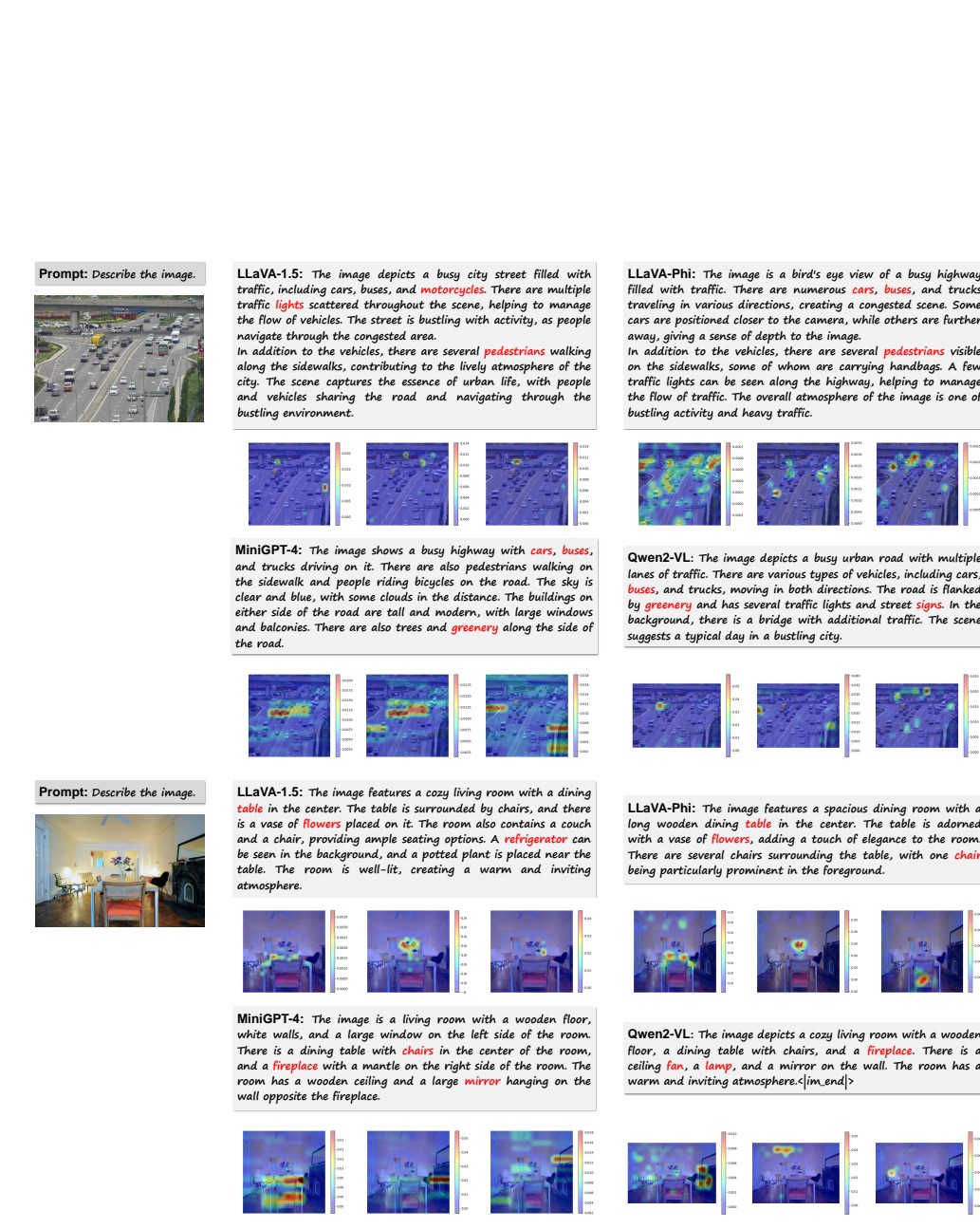

Figure 11: Additional visualization results for four different LVLMs using VaLSe.

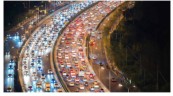

**Text prompts:** Describe this photo in detail.

**LLaVA:** The image depicts a busy highway at night, filled with heavy traffic. Numerous cars are driving in both directions, occupying the entire width of the road. There are also a few trucks among the traffic, adding to the congestion. In addition to the vehicles, there **are several pedestrians** visible in the scene, some of them carrying backpacks. They are scattered throughout the highway, with some close to the edge of the road and others further away. The overall atmosphere of the image is bustling and hectic, as people and vehicles navigate the busy highway at night.

**VaLSe:** The image depicts a busy highway filled with heavy traffic, both in the foreground and background. There are numerous cars, trucks, and buses driving on the highway, creating a sense of congestion. The traffic appears to be moving in both directions, with some vehicles closer to the foreground and others further back in the scene. The highway is surrounded by trees, adding a touch of nature to the urban landscape. The traffic is so dense that the highway appears to be a tunnel, with the vehicles moving in a continuous stream. The scene captures the essence of a bustling city and its transportation system.

----------------------------------------------------------------------------------------

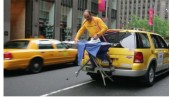

**Text prompts:** Describe this photo in detail.

**LLaVA :** The image features a man standing on the back of a yellow taxi cab, holding a blue shirt on a clothesline. The taxi is driving down a busy street, surrounded by other vehicles. There are two cars in the scene, one on the left side of the taxi and another on the right side. In addition to the taxi, there are **two trucks** visible in the image. One truck is located on the left side of the taxi, and the other truck is on the right side. The busy street and the presence of multiple vehicles suggest that the man is likely trying to dry his clothes in a public space.

**VaLSe:** The image captures a man standing on the back of a yellow taxi cab, holding a blue shirt in his hands. The taxi is driving down a busy city street, with other vehicles such as a truck and a car visible in the background. The man appears to be ironing the shirt while standing on the back of the taxi, possibly as a unique way to save space or time.

----------------------------------------------------------------------------------------

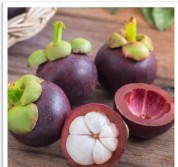

**Text prompts:** Describe this photo in detail.

**LLaVA:** The image features a wooden table **with a basket of fruit**, including **a variety of fruits** such as apples, oranges, and bananas. There are three apples on the table, with one placed near the left side, another in the middle, and the third one on the right side. In addition to the apples, there are two oranges on the table, one located in the middle and the other towards the right side. A banana can be seen on the left side of the table, and a cluster of grapes is also present, with some grapes placed near the center and others on the right side of the table. The assortment of fruits creates a colorful and appetizing display.

**VaLSe:** The image features a wooden table with four dark purple fruits, possibly pomegranates, cut in half. The fruits are placed on the table, with one of them being cut in half and the other three still intact. The cut fruit reveals the white interior, which contrasts with the dark purple exterior. The table appears to be a dining table, and the fruits are placed on it, ready to be enjoyed.

----------------------------------------------------------------------------------------

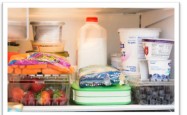

**Text prompts:** What is the brand of the yogurt flavored with blueberry?

**LLaVA:** The brand of the yogurt flavored with blueberry is **Chobani**.

**VaLSe:** The brand of the yogurt flavored with blueberry is Fage.

**Text prompts:** Is there any strawberry-flavored yogurt in the fridge?

**Qwen:** Yes, there **is strawberry-flavored yogurt** in the fridge.

**VaLSe:** There are strawberries in the left part of the fridge.

----------------------------------------------------------------------------------------

**Text prompts:** Give a detailed description of this meme.

**LLaVA:** The image is a comic strip featuring a man standing in front of a white board with a graph on it. The man is pointing at the graph, which appears to be a representation of a stacked layers model. The man is wearing a funny hat, and he seems to be explaining the concept to others. There are **several people** in the scene, with one person standing close to the man and others further away. The comic strip is divided into two panels, with the first panel showing the man pointing at the graph and the second panel showing the same scene but with **the man pointing in the opposite direction**.

**VaLSe:** The image consists of two separate panels, each featuring a cartoon man with a funny facial expression, likely representing a frustrated or angry person. In the first panel, the man is pointing at a graph, while in the second panel, he is pointing at a different graph. The man's facial expression and body language suggest that he is frustrated or upset about the graphs. In both panels, there are graphs or diagrams displayed, possibly representing data or information. The man's presence and the graphs suggest that he is discussing or analyzing the data.

Figure 12: Cases of VaLSe on LLaVA-Bench.

