# OpenReview forum: "Seeing It or Not? Interpretable Vision-aware Latent Steering to Mitigate Object Hallucinations"
_ICLR.cc/2026/Conference — ICLR 2026 Conference Withdrawn Submission_

### Official Review · Reviewer_QzA8 · 2025-10-29

**Soundness:** 3
**Presentation:** 3
**Contribution:** 3
**Rating:** 4
**Confidence:** 4

**Summary:**

The paper tackles object hallucination (OH) in large vision-language models (LVLMs), a problem where models generate object words not grounded in the image. The authors propose VaLSe (Vision‑aware Latent Steering), a framework that combines interpretability and mitigation. VaLSe first models the influence of visual inputs on each output token by computing visual token contribution maps using gradient‑weighted attention propagation and removes spurious “artifact” activations by contrasting with a system token. It identifies visual‑sensitive tokens via a log‑likelihood ratio (LLR) between predictions with the true image and with a noise image. Using these maps, the authors create positive and negative sample pairs by masking irrelevant regions and compute a latent steering direction via PCA over differences in internal activations. This steering vector is added to latent states during inference to shift the model toward semantically relevant content and suppress hallucinations. Experiments on multiple benchmarks and LVLMs show that VaLSe reduces hallucination metrics (CS/CI) and often improves F1 without compromising BLEU or response length.

**Strengths:**

S1. The framework systematically includes visual‑sensitive token selection, contribution map generation with artifact removal, and steering via PCA.

S2. Extensive experiments across models and benchmarks demonstrate consistent reductions in hallucination metrics without harming general performance.

S3. The paper is generally well‑structured.

**Weaknesses:**

W1. VaLSe requires additional forward passes to compute LLR for each token and gradient‑based contribution maps, followed by PCA on latent differences. While the method is training‑free, inference‑time cost may be high for long responses or high‑resolution images. A clearer discussion of computational cost and optimizations would be useful.

W2. The LLR threshold alpha and the number of masked regions are tuned manually. Although ablations suggest robustness, automatic selection strategies or adaptive thresholds across domains are not explored. VaLSe might require per‑model calibration, limiting out‑of‑the‑box usability.

W3. The case study shows that VaLSe can steer attention toward the wrong region in the presence of typographic deception. More thorough analysis is needed on failure modes or when visual priors are ambiguous.

**Questions:**

Q1. What is the approximate computational overhead (in terms of FLOPs or latency) of VaLSe compared with standard inference? Could pruning the number of selected tokens or approximating contribution maps reduce cost while retaining effectiveness?

Q2. Have you considered adaptive methods to set the LLR threshold alpha (e.g., based on the distribution of LLR values within a sample) instead of a fixed value? Would such methods generalize better across different models and datasets?

Q3. VaLSe highlights limitations in existing metrics. Have you explored using VaLSe’s visual maps to build improved evaluation benchmarks or metrics that better distinguish true hallucinations from unannotated but correct objects?

---

### Official Review · Reviewer_Tymj · 2025-10-30

**Soundness:** 2
**Presentation:** 3
**Contribution:** 2
**Rating:** 4
**Confidence:** 4

**Summary:**

The paper introduces VaLSe, a training-free, interpretation-driven framework for mitigating object hallucination in large vision–language models. VaLSe first derives token-level visual contribution maps by propagating gradient-weighted attention and suppressing spurious activations via system-token contrast, thereby isolating image-grounded evidence. It then constructs masked/unmasked feature pairs and performs PCA to obtain a vision-aware latent steering direction, which is injected into intermediate LLM states at inference to bias decoding toward visually supported content. Evaluations on multiple hallucination benchmarks indicate reductions in hallucinated objects with limited degradation of general multimodal performance, while qualitative analyses highlight systematic mislabeling in existing OH metrics.

**Strengths:**

1. The writing is organized and clear, with the core idea of moving from interpretation to mitigation.
2. The paper usefully points out that existing OH metrics can mislabel visually grounded tokens and backs this with qualitative examples.

**Weaknesses:**

1. The proposed interpretation–then–steering pipeline is noticeably heavier than decoding-only hallucination controls. To obtain token-wise visual contribution maps, VaLSe has to (i) identify visual-sensitive tokens via an image/noise comparison and (ii) for each selected token, run a Grad-CAM–style gradient-weighted propagation over all attention layers, which already implies at least one backward-style pass on top of the normal generation. On top of that, the steering direction is computed from paired positive/negative samples that both require forward features from the MLP layers. As a result, the paper would benefit from runtime/memory profiling and a discussion of efficiency. In deployment settings, especially when only small gains are observed (e.g. on POPE with LLaVA-1.5 the improvement is about 1%), it is unclear whether this extra gradient computation and double inference is preferable to simpler decoding-level manipulation.
2.  The paper provides comparison with existing hallucination mitigation methods mainly on CHAIR, where VaLSe shows gains. On other hallucination benchmarks such as AMBER, POPE, MMHal and MMVP, the results are reported only as base model versus base model plus VaLSe, so it is not possible to see whether VaLSe still outperforms decoding-level methods under the same setting. Since the method is sensitive to vision token compression and architecture choices, evaluating on more LVLMs and running the same set of mitigation baselines on multiple benchmarks would make the claim of general applicability stronger.

**Questions:**

1. How expensive is VaLSe end to end compared with decoding-time hallucination mitigation methods such as HALC, VCD or OPERA?
2. How robust are the improvements to prompt paraphrasing and option shuffling?
3. Can you run the same set of baselines on AMBER, POPE and MMHal, etc. so we can see whether VaLSe is still preferable once compute is taken into account?

---

### Official Review · Reviewer_8eAf · 2025-10-31

**Soundness:** 2
**Presentation:** 3
**Contribution:** 2
**Rating:** 4
**Confidence:** 3

**Summary:**

This paper studies object hallucination (OH) in large vision-language models (LVLMs), where models generate outputs inconsistent with visual inputs. It proposes VaLSe, a Vision-aware Latent Steering framework that first interprets how visual information contributes to each output token, then mitigates hallucination by steering latent representations toward semantically relevant content. VaLSe produces visual contribution maps that show the model’s attention and decision focus, helping both interpretation and correction. Experiments show that VaLSe reduces hallucination and improves model robustness across benchmarks. The paper also points out weaknesses in current OH evaluation metrics and calls for more interpretable and visually grounded benchmarks.

**Strengths:**

Strengths
1. The motivation is strong and well explained, addressing an important issue of object hallucination in LVLMs.
2. The proposed method, VaLSe, is conceptually clear and combines interpretability with mitigation in a meaningful way.
3. The approach provides useful visual contribution maps that make the model’s reasoning more transparent.

**Weaknesses:**

Weaknesses
1. The reliability of the visual contribution maps is not fully verified, and the connection between visualization and real causal influence may be weak.
2. The method may add extra computation and inference time because of its two-stage process.
3. The effect of latent space steering may depend heavily on which layers or parameters are adjusted, and this sensitivity is not deeply analyzed.

**Questions:**

Questions
1. How reliable are the visual contribution maps in reflecting true causal visual influence?
2. What is the computational cost of VaLSe compared to standard LVLM inference?
3. How sensitive is the method to the choice of layers or strength of latent steering?

---

### Official Review · Reviewer_9xjT · 2025-10-31

**Soundness:** 2
**Presentation:** 2
**Contribution:** 2
**Rating:** 4
**Confidence:** 5

**Summary:**

The paper addresses object hallucination by introducing a novel latent steering method called VaLSe. VaLSe is composed of the following components.

1. Visual Token Contribution Maps: VaLSe identifies the set of visual tokens by thresholding the LLR value. The tokens are then used to get the contribution map by Eq. (5).
2. Sample Generation: Positive and negative samples are needed to obtain the steering direction. The masked samples are generated from the contribution map. These samples are used as the negative samples because the visual region is masked.
3. Latent Steering: The direction is obtained by computing the difference between the positive and negative samples and using PCA. During inference, the learned direction is used to shift the original latent representation toward a non-hallucinated latent space.

The paper validates the proposed method on diverse hallucination benchmarks, including POPE and CHAIR. Also, the paper shows that the general capabilities do not degrade.

**Strengths:**

1. The proposed method does not decrease the general task performance. It is essential to address object hallucination while preserving the general capabilities of LVLMs.
2. VTI is the closest related work to the proposed method, VaLSe. In VTI, they generate the positive and negative samples of images and texts. Then, they compute the steering direction using PCA. This direction is used for test-time inference through intervention. The proposed method advances the generation modules of VTI. The design choice is reasonable, as it leverages the model's response to guide sample generation.

**Weaknesses:**

1. In Section 3.3, the paper gives an analysis to understand the model's behavior under perturbed inputs. However, the derivation is hard to follow.
	- The used notation lacks clear definitions, making it challenging to understand the mathematical reasoning.
	- The claim that the attention matrix of ideal noise is zero requires further justification.
2. ValSe requires the dataset for learned direction (Eq. 6). However, the optimization detail is ambiguous. Specifically, it is unclear how the hyperparameters $\lambda$ and $\alpha$ are set.
3. VTI is the closest related work. Therefore, a detailed comparison is necessary to highlight the proposed method. Additionally, the table currently reports only the performance of VTI-vision; the full VTI performance should be included.
4. Other steering methods, such as Nullu, should also be discussed and reported, as the proposed approach falls within this research approach.
5. The performance of different methods on other hallucination benchmarks remains unreported.

**Questions:**

1. What if bounding boxes are used instead of contribution maps or sample generation?

---

### Note · Authors · 2025-11-17

I have read and agree with the venue's withdrawal policy on behalf of myself and my co-authors.